# Usefulness of Heat Map Explanations for Deep-Learning-Based Electrocardiogram Analysis

**DOI:** 10.3390/diagnostics13142345

**Published:** 2023-07-11

**Authors:** Andrea M. Storås, Ole Emil Andersen, Sam Lockhart, Roman Thielemann, Filip Gnesin, Vajira Thambawita, Steven A. Hicks, Jørgen K. Kanters, Inga Strümke, Pål Halvorsen, Michael A. Riegler

**Affiliations:** 1Department of Holistic Systems, Simula Metropolitan Center for Digital Engineering, 0167 Oslo, Norway; 2Department of Computer Science, Oslo Metropolitan University, 0130 Oslo, Norway; 3Department of Public Health, Aarhus University, 8000 Aarhus, Denmark; 4Steno Diabetes Center, Aarhus University, 8000 Aarhus, Denmark; 5Wellcome Trust-Medical Research Council Institute of Metabolic Science, University of Cambridge, Cambridge CB2 0QQ, UK; 6Novo Nordisk Foundation Center for Basic Metabolic Research, University of Copenhagen, 2200 Copenhagen, Denmark; 7Department of Cardiology, North Zealand Hospital, 3400 Hillerød, Denmark; 8Department of Biomedical Sciences, University of Copenhagen, 2200 Copenhagen, Denmark; 9Department of Computer Science, Norwegian University of Science and Technology, 7491 Trondheim, Norway; 10Department of Computer Science, UiT The Arctic University of Norway, 9037 Tromsø, Norway

**Keywords:** explainable artificial intelligence, electrocardiograms, heat maps

## Abstract

Deep neural networks are complex machine learning models that have shown promising results in analyzing high-dimensional data such as those collected from medical examinations. Such models have the potential to provide fast and accurate medical diagnoses. However, the high complexity makes deep neural networks and their predictions difficult to understand. Providing model explanations can be a way of increasing the understanding of “black box” models and building trust. In this work, we applied transfer learning to develop a deep neural network to predict sex from electrocardiograms. Using the visual explanation method Grad-CAM, heat maps were generated from the model in order to understand how it makes predictions. To evaluate the usefulness of the heat maps and determine if the heat maps identified electrocardiogram features that could be recognized to discriminate sex, medical doctors provided feedback. Based on the feedback, we concluded that, in our setting, this mode of explainable artificial intelligence does not provide meaningful information to medical doctors and is not useful in the clinic. Our results indicate that improved explanation techniques that are tailored to medical data should be developed before deep neural networks can be applied in the clinic for diagnostic purposes.

## 1. Introduction

Electrocardiograms (ECGs) are widely used in hospitals and clinics to evaluate the electrical currents in the heart and discover anomalies such as Myocardial Infarction (MI). The method provides high-resolution time series data and is fast, inexpensive, and easily accessible, making ECGs an obvious target for Artificial Intelligence (AI). Machine Learning (ML) is a subfield of AI that learns from data. The usefulness of ML to extract new information from ECGs has been shown by earlier research that predicted sex from ECGs [1], a phenotype medical doctors are unable to infer from reading ECGs. There is also a high potential for exploiting the power of ML to develop systems for the automatic medical diagnosis of patients. Earlier research has, e.g., shown that ML models are able to accurately diagnose atrial fibrillation, hypertrophic cardiomyopathy, and long QT syndrome from ECGs [2,3,4]. However, understanding what parts of the input data ML models use to make predictions is often challenging. This is especially true regarding complex models such as deep neural networks. Explainable Artificial Intelligence (XAI) aims to make ML models more interpretable by providing explanations about the models and their predictions. Explaining a model and its outputs can increase the end-users’ trust in the model. In the medical field, the lack of interpretability of ML models has been identified as an important challenge that can limit the implementation of such models in the clinic [5,6]. From a clinical perspective, it is, therefore, important to successfully explain the model’s outputs to healthcare personnel. Moreover, understanding the model’s predictions can provide insight into the domain of interest since the model might detect relationships that are not obvious to humans [1].

Prior work evaluating heat maps for explaining diagnostic models in the fields of radiology and ophthalmology concluded that the explanations did not highlight areas of relevance for medical doctors [7,8]. On the other hand, heat maps did focus on a specific feature in ECGs when a deep neural network was developed to distinguish sex, providing new medical knowledge to cardiologists [1]. Due to diverging results, the aim of this paper was to further explore whether visual XAI methods can increase our understanding of “black box” ML models for a medical application. More specifically, we investigate the usefulness of heat maps in the clinic for interpreting sex differences in ECGs, inspired by the work by Hicks et al. [1]. It is increasingly recognized that females have higher rates of misdiagnosis of MI relative to males [9] and suffer longer delays before treatment [10]. While the observed sex differences likely relate predominantly to differences in illness presentation, we reasoned that understanding sexual dimorphism in ECG presentation of MI could be useful in defining sex-specific adverse ECG features. We applied transfer learning of a deep neural network, originally developed on data from two population studies [11,12], to predict a patient’s sex from ECG signals. The model was fine-tuned on data from the PTB-XL dataset [13,14]. Furthermore, heat maps were created using an explanation method called Gradient-Weighted Class Activation Mapping (Grad-CAM) to explain model predictions for normal and MI ECGs [15]. Practicing physicians evaluated the heat maps, concluding that the heat maps were not clinically useful and did not highlight consistent waveforms in the ECGs. Instead, the heat maps increased the skepticism toward the model. Before applying deep neural networks for ECG-based medical diagnoses in the clinic, we, therefore, recommend developing explanation methods that are tailored to medical data analysis and towards medical practitioners’ needs.

The main contributions of our work are:We assessed the usefulness of Grad-CAM-based heat maps as a tool for interpreting deep neural networks’ decision-making processes in analyzing ECGs, offering a critical evaluation of the utility of current XAI methodologies in clinical settings.Our findings highlight the shortcomings of the existing explanation methods, specifically in their failure to provide meaningful or in clinical practice applicable insights to medical practitioners.We emphasize the need for critical evaluations of existing methods and for improvements and advancements in XAI techniques before they can be effectively utilized for clinical practice.We argue for the development of more holistic XAI methods that cover all aspects of a model’s functioning, from its development to its predictions (batch or real-time).We underline the importance of interdisciplinary collaboration in creating model explanations that are genuinely useful to healthcare professionals, ensuring that these methods meet the unique needs of medical professionals.

The rest of the paper is structured as follows: The dataset and method are described in Section 2. The results are presented in Section 3 and discussed in Section 4. Finally, conclusions are drawn in Section 5.

## 2. Materials and Methods

### 2.1. Dataset

We obtained raw signal data from 21,837 12-lead ECGs from 18,885 patients (some patients contributed several ECGs) collected using devices from Schiller AG between 1989 and 1996 from the PTB-XL dataset [13,14] publicly accessible at Physionet [16]. The dataset has been described in detail previously [13]. The ECGs lasted for 10 s each and were recorded using a sampling frequency of 500 Hz [14]; see also Figure 1. Patient characteristics, including sex, age, body weight, and cardiac diagnoses, are also available. The dataset is balanced with regard to sex. Five diagnostic superclasses are included: normal, MI, ST-segment and T-wave changes, conduction disturbances, and hypertrophy. Normal ECGs and MI are the largest superclasses [13].

### 2.2. Deep Neural Network

Generative Adversarial Networks (GANs) consist of two ML models: a generator and a discriminator. The generator generates fake data, which mimic the observations in the training dataset. The discriminator is trained to distinguish real data in the training dataset from fake data produced by the generator. The generator and discriminator are trained together by competing against each other. The generator tries to fool the discriminator by creating increasingly more realistic data, while the discriminator tries to disclose the generator [17]. Because the discriminator is a binary classifier, it can be fine-tuned to perform other classification tasks.

Because the focus of this study was to evaluate model explanations in clinical practice rather than developing novel neural network architectures, we applied transfer learning using the discriminator of a previously developed GAN called Pulse2Pulse [18]. The GAN was trained on data from two population studies [11,12] to produce normal ECGs. Pulse2Pulse can generate fake ECGs with similar intervals and amplitudes as real ECGs [18]. We transferred the weights from the discriminator of the GAN after it had been trained for 2500 epochs, at which point, the GAN reached its highest performance in the original paper [18]. Originally, the discriminator classified ECGs as real or fake. In our experiments, we changed the discriminator to predict sex by fine-tuning it on the training and validation data from the PTB-XL dataset. Its architecture is outlined in Figure 2. Transfer learning is an attractive tool, especially in cases with little training data [19]. Similar approaches have been described for damage detection in concrete structures [20] and stain-free cell classification [21]. To our knowledge, this is the first time transfer learning has been applied to a GAN discriminator for classification in cardiology.

Due to computational demand, we used 5000 and 200 samples for training and validation, respectively. These samples were randomly picked from the recommended training and validation parts of the PTB-XL dataset, eliminating the risk of selection bias. The model was fine-tuned for 1000 epochs on the selected training and validation samples, and the model weights from the epoch with the highest accuracy on the validation samples were saved. The full test set was used to evaluate the final model. Because a single patient could contribute with more than one observation (ECG) to the dataset [13], we ensured that all ECGs from the same patient were placed in either the training, validation, or test set. This way of splitting the data on the patient level was performed to avoid data leakage. For comparing model performance on MI ECGs and normal ECGs, samples annotated as either MI or normal were extracted from the test set and analyzed as two separate sub-test sets. Subjects included in these subsets were also present in the full test set. To avoid ECGs with empty or contradicting diagnostic superclasses, only subjects with one registered diagnostic superclass were included in the additional sub-test sets. The distributions of males and females in the data used to train and test the model are provided in Table 1.

### 2.3. Model Explanations

There are several ways to explain ML models. Intrinsic explanations aim to explain the inner workings of a model by using the model’s internal weights. Grad-CAM [15] is among the most-popular visual intrinsic explanation methods. Since it also showed successful results in explaining sex predictions from ECGs in earlier work [1], Grad-CAM was a natural choice of explanation method in this study. The method provides heat maps that highlight the areas in an image where the activations in the model are strongest given a certain prediction. Consequently, the end-user can investigate what regions in the image the model reacts to when predicting what the image represents. Even though ECGs are time series data, they can be analyzed using the standard 10 s rhythm strip containing registered electrical signals captured over eight leads. We adopted this approach, making it possible to apply Grad-CAM to our model and data. For heat map generation, a modified version of the PyTorch Grad-CAM package was applied (https://github.com/jacobgil/pytorch-grad-cam, accessed on 6 July 2023). Heat maps were extracted from the penultimate convolutional layer in the neural network to obtain adequate resolution for the heat maps. The resulting heat maps had the time dimension on the x-axis and the amplitude of the ECG signals on the y-axis.

### 2.4. Feedback from Physicians

To evaluate the heat maps as an explanation method and determine if our methodology identified ECG features that could be recognized to discriminate sex, two practicing physicians examined the heat maps with ECG waveforms overlaid. The physicians were located in the United Kingdom and Denmark, respectively, and had experience in general practice, general internal medicine, diabetes, and endocrinology. The feedback from the physicians was structured as an interview, where the physicians inspected the heat maps and spoke out about what they thought. They also asked questions if something was unclear and suggested improvements. The interviews were recorded for future reference after acquiring oral informed consent.

Furthermore, some selected heat maps were qualitatively evaluated for closer investigation of the quality. The heat maps were chosen purposefully, selecting the 15 heat maps from normal male and female ECGs that were classified with the highest certainty by the model. All eight leads that the model was trained on (Leads I, II, and V1–V6) were included in the heat maps. Each physician evaluated the heat maps independently, blinded to sex, and provided a written summary of their impressions. The results were discussed in plenary before a common decision was made, inspired by the Delphi method [22].

### 2.5. Technical Details

All programming was performed using Python Version 3.8.11 and the PyTorch library Version 1.9.1 for the deep neural network. The experiments were performed on an NVIDIA DGX-2 server consisting of 16 NVIDIA Tesla V100 GPUs with 32 GB memory each. The source code is publicly available (https://github.com/AndreaStoraas/UsefulnessECG_Heatmaps.git, accessed on 6 July 2023).

## 3. Results

### 3.1. Model Performance

The model performance was measured based on the model’s ability to separate males and females ECGs. Figure 3 shows the accuracies of the final model on the training, validation, and test sets. The model achieved an accuracy of 81% on the validation set and 74% on the hold-out test set. Moreover, the model performed better on men than women. For additional insight into the error distribution, the confusion matrix for the test set is provided in Figure 4. When comparing the model performance on the subsets of the test set containing normal ECGs and MI ECGs, the accuracy dropped from 80% to 71%. Again, the model performed better on men than women.

### 3.2. Heat Maps

To explore which features of ECGs drove sex discrimination by the model, we created heat maps using observations in the normal and MI sub-test sets that were correctly classified by the model. In total, 919 heat maps were created and analyzed. The heat map colors ranged from red to blue, where red represents the most-important areas and blue represents the least-important areas. During heat map inspection, we found patterns of dark-blue heat maps, indicating low model importance, and brighter or red heat maps, indicating higher model importance. A general finding was that females typically had dark-blue heat maps, while males had heat maps that were more evenly distributed between blue and red. More specifically, 93.3% of the heat maps from females were dark, while the corresponding number for heat maps from males was 60%. Examples of heat maps of different color patterns are provided in Figure 5. The ECG signals from Lead II are overlaid.

### 3.3. Feedback from Physicians

Two practicing physicians independently reviewed a sample of heat maps in an attempt to identify features in ECG waveforms that our model used for sex prediction and to evaluate the heat maps’ usefulness for explaining the model predictions. Unfortunately, neither physician could identify features of the ECG waveform that consistently corresponded to the highlighted areas of the heat map, which the model utilized for discrimination.

Most of the heat maps did not highlight the same part of each QRS complex, such as the downslope of the R-wave. This was found to be counterintuitive by the physicians because they usually interpret QRS complexes separately. The physicians expected the model to react to the same part of each QRS complex, making a repeating pattern of red areas in the heat maps. Since this was typically not the case, it was difficult to link the high-attention areas to specific features of an ECG. Moreover, the physicians’ confidence in the deep neural network decreased after the heat map evaluation.

Some ECGs had quite bad quality and were challenging for medical doctors to interpret. The physicians suspected that the bad-quality ECGs might explain why the heat maps were uninformative since the model can learn and react to noise in the data.

The extracted heat maps for female individuals were typically dark-blue (indicating that Grad-CAM did not find characteristic features in the females), while this was not the case for males. The physicians suggested that the model learned characteristic patterns for males. If these patterns were absent, the model simply classified the ECG as female.

During the interviews, the physicians asked for the predicted probabilities and the data distribution. We, therefore, extracted and inspected the predicted probabilities together with the heat maps; see also Figure 5.

The physician consensus was that our XAI approach is not useful for identifying ECG features that can highlight novel sexually dimorphic electrocardiographic features to physicians.

## 4. Discussion

The physicians could not define any clinically meaningful ECG waveforms that the model consistently reacted to. As such, it is unlikely that the heat maps will be helpful in highlighting sexually dimorphic ECG features for clinical practice. More specifically, the physicians found it difficult to understand the heat-map-generation process and thought that the model reacted to areas in the ECGs that doctors would not react to. Even though medical doctors do not know how to infer sex from ECGs, earlier findings suggest the downslope of the R-wave to be an important feature for sex classification [1]. Despite applying the same heat map method in this study, our heat maps did not consistently highlight these ECG segments. Because the model interpreted 10 s-long ECGs, the attention was distributed over a larger period of time, not focusing on one QRS complex, which may result in smearing the attention out. Training the model on the median ECGs instead might make the heat map evaluation easier for medical doctors. Median ECGs aggregate the periodic signals and contain less noise. Each heat map is then restricted to a single QRS complex, which might be more informative to physicians. This way of aggregating periodic signals can be a useful preprocessing method when explanations are given as heat maps, also beyond ECG analysis. In this study, median ECGs were not applied due to the nature of the available data and because we applied transfer learning from an earlier model, which was developed on 10 s ECGs.

Heat maps have several drawbacks as explanation methods. First, they depend on which layer in the neural network they are extracted from. For Grad-CAM, the resolution of the heat map is similar to the size of the convolutional feature map [15]. The last convolutional layers are typically chosen because they capture higher-level details in the input data while also containing detailed spatial information [15]. In our model, the last convolutional layer had only five neurons, resulting in heat maps of low resolution. For improved resolution, heat maps were extracted from the second to last convolutional layer. Furthermore, heat maps explain single predictions and are less suitable for providing global explanations about how the model works on the entire dataset. For future work, different techniques for aggregating heat maps from several predictions will be tested.

Prior research supports our findings, showing that model explanations do not always meet the needs of medical doctors [5,23]. One study explaining a deep neural network predicting Alzheimer’s disease found heat maps not to produce medical imaging biomarkers that could be interpreted by humans [24]. For skin cancer detection, researchers found Grad-CAM and kernelSHAP [25] not to provide heat maps with clinically meaningful information [26]. Further on, two studies thoroughly evaluated several heat map methods for interpreting X-rays analyzed using deep neural networks [7,27]. They concluded that the heat maps were unable to generate consistent explanations that also highlighted the regions of interest [7] and warned against applying the current heat map methods for explaining deep neural networks in medical imaging [27]. Medical images can be quite different from natural images, and deep neural networks are sometimes distracted by parts of the data that seem irrelevant for the task [28]. Consequently, future work should explore other XAI techniques than heat maps to explain ML models for ECG analysis. Moreover, we experienced that the physicians asked for the model uncertainty and the dataset distributions of sex and diagnoses. Including this additional information can, therefore, be useful when explaining and evaluating ML models for medical applications and increase the doctors’ trust in the models.

One concern among the physicians was the quality of the ECGs used for model development. The data included in the PTB-XL dataset were collected more than 20 years ago, between 1989 and 1996 [13]. At that time, the error rate for recording data using medical devices was higher than today. Input data of bad quality will negatively affect the generated heat maps and further affect the physicians’ heat map evaluations. However, our model’s performance was acceptable, and other deep neural networks developed on the same dataset for the same task achieved high accuracies (>84%) [29]. Therefore, the data seem to be of sufficiently high quality for training ML models. The physicians’ negative impression of the heat maps was most likely due to the applied explanation method and not the neural network. Interestingly, the trust in the model decreased after showing the heat maps to practicing physicians. Rather than being a useful tool for enhanced implementation of ML systems, we found that explanations can, in some cases, increase the skepticism toward the model. Taken together, our results underline the importance of tailoring the explanation method to the specific use case and the end-users.

Our ML model achieved higher performance in classifying normal ECGs than MI ECGs. One possible reason is that the majority of the ECGs in the training data were annotated as normal, meaning that the model will be less familiar with MI ECGs. Because the original discriminator was only trained on normal ECGs, this could also contribute to the observed results. Nevertheless, the most-likely reason is disturbances in the ECGs caused by MI, masking patterns that are used to distinguish between the sexes. Furthermore, the model was better at classifying men than women, regardless of diagnosis. As shown in Table 1, the sex distribution of the dataset was balanced, so it is unlikely that the observed performance differences were caused by an overrepresentation of men. Indeed, ECGs differ between the sexes [30,31], and female ECGs are more difficult to interpret than male ECGs. Perhaps male ECGs include patterns that were recognized more easily by the model.

Our deep neural network was trained to infer sex from ECGs. While sex prediction is not something that would be used in a clinical setting, we believe our evaluation of the heat map explanations can be generalized to other deep-learning-based ECG analyses, including the diagnosis of various cardiac conditions. Further research should, therefore, look into how to explain deep neural networks for medical diagnosis in a way that meets the needs of healthcare personnel.

A limitation of the current work is that we qualitatively evaluated one heat map method, namely Grad-CAM, for explaining one deep neural network. An extended study that both qualitatively and objectively evaluates a larger number of heat map methods on several deep neural networks for ECG analysis has already been initiated by our research group. It will be interesting to compare the findings from the presented work with similar evaluations of other models and heat map methods.

## 5. Conclusions

This study investigated the usefulness of heat maps for explaining a deep neural network predicting sex from ECGs. The results showed that the explanations were not found useful by the domain experts and that the heat maps could not be applied to obtain new medical knowledge. Our findings indicate that it is necessary to improve the existing methods before they are ready to explain deep neural networks for diagnostic purposes in the clinic. More effort should be made to develop holistic explanations that cover all aspects of the model, from development to real-time predictions. Moreover, interdisciplinary collaboration is essential to ensure that model explanations are useful for healthcare personnel.

## Figures and Tables

**Figure 1 diagnostics-13-02345-f001:**
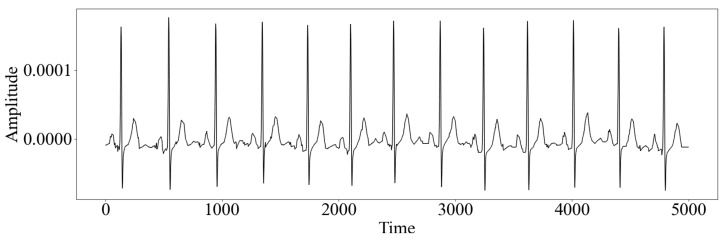
Example of an Electrocardiogram (ECG) from the PTB-XL dataset, showing Lead II. Time is on the x-axis, and amplitude is on the y-axis. The ECGs were sampled at a frequency of 500 Hz per second. Each ECG lasted for 10 s, giving 5000 measurements per lead.

**Figure 2 diagnostics-13-02345-f002:**
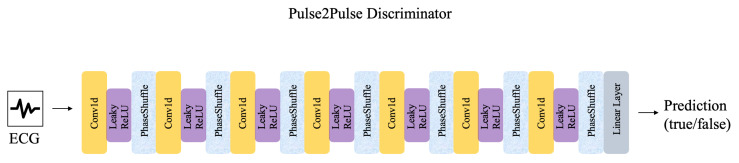
Architecture of the Pulse2Pulse discriminator, which was applied for transfer learning in the current study. Leaky ReLU is the activation function. Abbreviations: Conv1d = one-dimensional Convolutional layer, PhaseShuffle = Phase Shuffling layer. The figure was inspired by the figure in [18].

**Figure 3 diagnostics-13-02345-f003:**
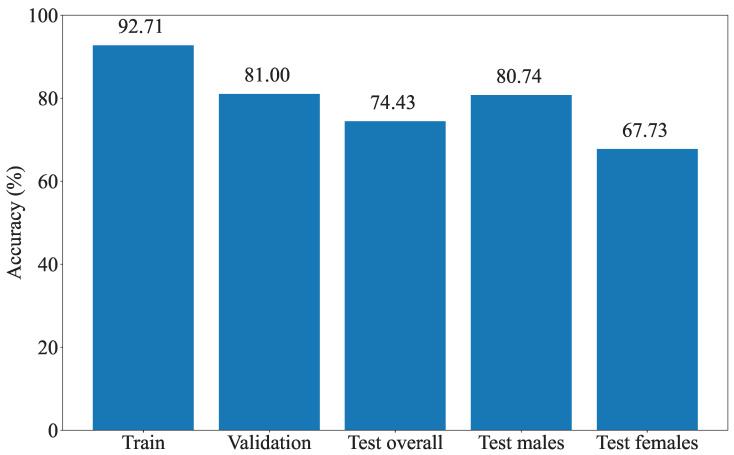
Performance of the Machine Learning (ML) model on the training, validation, and test sets. Separate accuracies for males and females in the test set are also included.

**Figure 4 diagnostics-13-02345-f004:**
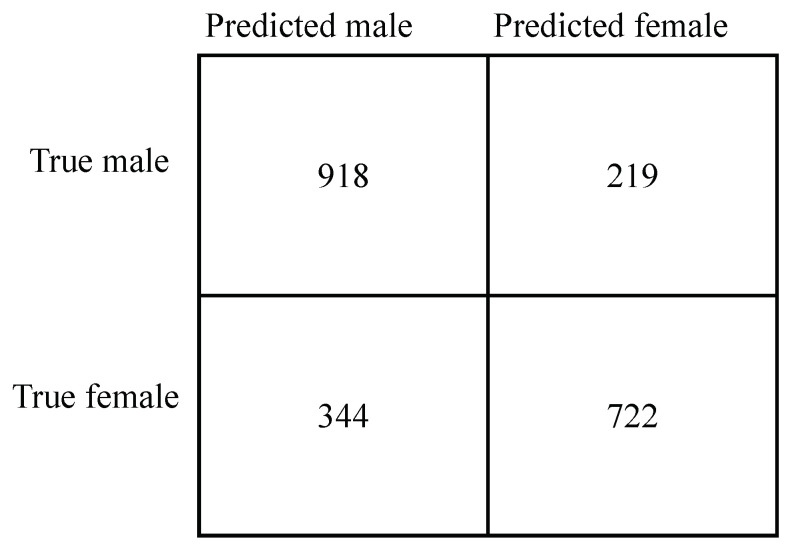
Confusion matrix for the final model when predicting sex on the full hold-out test set.

**Figure 5 diagnostics-13-02345-f005:**
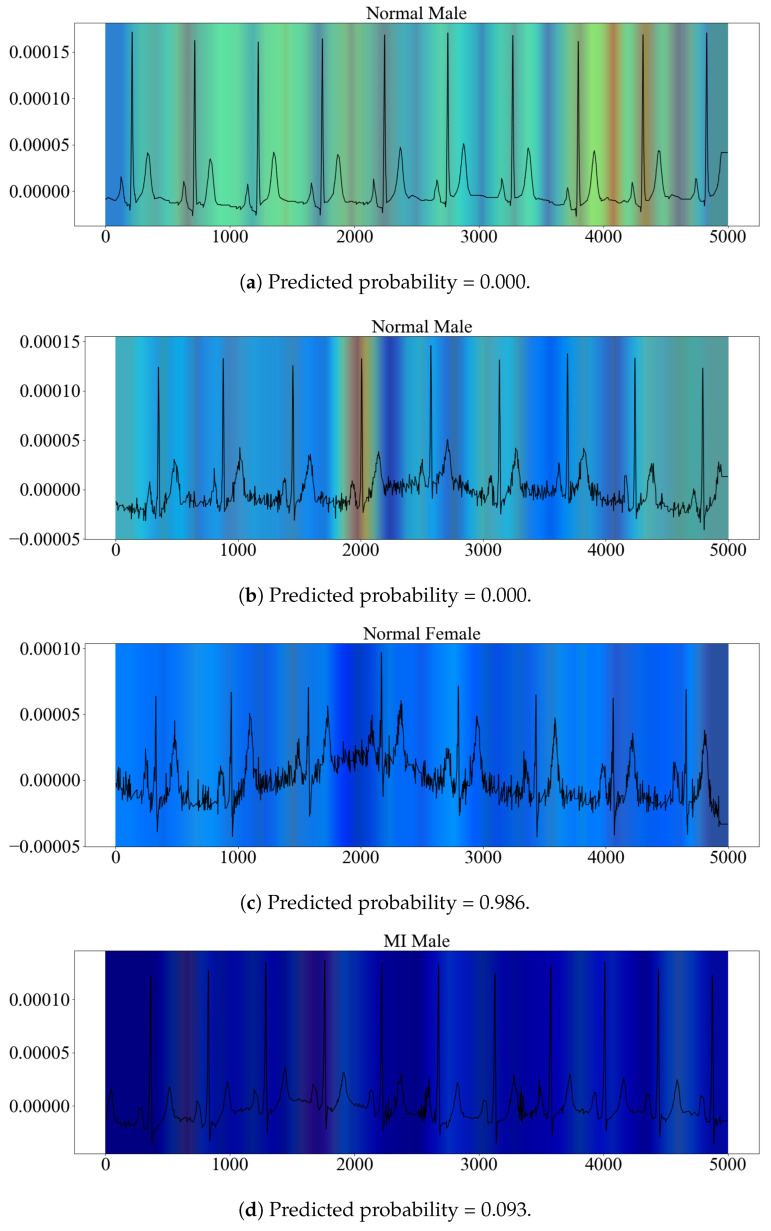
Example heat maps with overlaid ECG signals showing different color patterns. Corresponding predicted probabilities are provided below each example. For Subplots (**a**,**b**,**d**), values close to 0 indicate higher model confidence, while for Subplot (**c**), values close to 1 indicate higher confidence.

**Table 1 diagnostics-13-02345-t001:** Sex distributions in the training, validation, and test sets. Test normal and test Myocardial Infarction (MI) ECG are subsets of the full test set.

Dataset	Female	Male	Total
Training	2437	2563	5000
Validation	90	110	200
Test	1066	1137	2203
Test normal ECG	427	486	913
Test MI ECG	118	138	256

## Data Availability

The PTB-XL dataset [13,14] is publicly accessible at Physionet [16] through the following link: https://doi.org/10.13026/kfzx-aw45, accessed on 6 July 2023.

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
