# Peer review of "Usefulness of Heat Map Explanations for Deep-Learning-Based Electrocardiogram Analysis"

_diagnostics, 2023, doi:10.3390/diagnostics13142345_

Round 1

Reviewer 1 Report

I have reviewed the manuscript with Usefulness of Heatmap Explanations for Deep Learning-Based Electrocardiogram (ECG) Analysis title. In this study, the authors applied transfer learning to develop a deep neural network to predict sex from electrocardiograms. To evaluate the usefulness of the heatmaps and determine if the heatmaps identifed electrocardiogram features that could be recognized to discriminate sex, medical doctors provided feedback. According to me, this manuscript has been organized, presented, and contains some scientific results but it is need some corrections to be enough to be accept for publication in Diagnostics.

1. The dataset used in the article is very old (1989-1996). The medical measurement technologies used at that time were old systems and the error rates were higher than today's medical devices. 2.  The GAN ML model used in the article is a known model and there is no innovation and development in the model. Hybrid models could be developed (Deep learning- GAN etc.). 3. It is unclear why certain medical doctor were selected for feedback. Authors should provide more information about the selection process to ensure that the results are representative of the medical doctor population.

4. There isn’t study has been conducted regarding the limitations of this study. There isn’t detailed literature study. It is not clear why the authors wanted to estimate age from ECG.

5. The authors of the study as a result of this article was concluded that "does not provide meaningful information for medical doctors and is not clinically useful". As a result, the authors have nothing to gain! It has no contribution to medical diagnosis, literature or innovation.

Basedon the issues mentioned above, we cannot recommend this article for publication in its current form. However, with the addition of relevant details and further discussion of implications, this article could be strengthened to an appropriate level for acceptance.

Author Response

Thank you for your comments. Please see attachment.

Reviewer 2 Report

The paper proposed evaluated the usefulness of heatmap in deep learning based ECG analysis and proved that it is useless. However, the experiments have some unclear issues.

Comments-1: you said that “ECGs from a single patient were placed in either the train, validation, or test set to avoid data leakage”. Does it mean the training and test sets are selected with replacement?

Comments-2: in the sentence “samples annotated as either MI or normal were extracted from the test set and analyzed as two additional test sets”, do you mean a single instance is used twice as either MI or normal? What is the influence of such data handling? Positive or negative?

Comments-3: in table 1, what is the relation between the Test dataset, Test normal ECGS, and Test MI ECGs?

Round 2

Reviewer 1 Report

Thank you for the replies and corrections to the comments.